# DNA Methylation and Alternative Splicing Safeguard Genome and Transcriptome After a Retrotransposition Burst in *Arabidopsis thaliana*

**DOI:** 10.3390/ijms26104816

**Published:** 2025-05-17

**Authors:** Pavel Merkulov, Anastasiia Latypova, Kirill Tiurin, Melania Serganova, Ilya Kirov

**Affiliations:** 1All-Russia Research Institute of Agricultural Biotechnology, Timiryazevskaya Str. 42, 127550 Moscow, Russia; paulmerkulov97@gmail.com (P.M.); ana.a.latypova@gmail.com (A.L.); tiurin.kn@gmail.com (K.T.); melany02@mail.ru (M.S.); 2Moscow Center for Advanced Studies, Kulakova Str. 20, 123592 Moscow, Russia

**Keywords:** *ONSEN* retrotransposon, *Arabidopsis thaliana*, transposable elements, DNA methylation, alternative splicing, RdDM pathway, nanopore sequencing

## Abstract

Transposable elements (TEs) are major drivers of plant genome plasticity, but the immediate molecular consequences of new TE insertions remain poorly understood. In this study, we generated a wild-type *Arabidopsis thaliana* population with novel insertions of *ONSEN* retrotransposon to investigate early epigenomic and transcriptomic changes using whole-genome and cDNA nanopore sequencing. We found that novel *ONSEN* insertions were distributed non-randomly, with a strong preference for genic regions, particularly in chromatin enriched for H2A.Z, H3K27me3, and H3K4me2. Most full-length *ONSEN* insertions within genes were rapidly recognized and spliced out as new introns (intronization), thereby mitigating potential deleterious effects on transcript isoforms. In some cases, *ONSEN* insertions provided alternative transcription start or termination sites, generating novel transcript isoforms. Genome-wide methylation analysis revealed that new *ONSEN* copies were efficiently and precisely targeted by DNA methylation. Independently on the location of the original *ONSEN* element, the euchromatic and heterochromatic insertions display distinct methylation signatures, reflecting the action of different epigenetic pathways. In conclusion, our results demonstrate that DNA methylation and alternative splicing are effective control mechanisms safeguarding the plant genome and transcriptome integrity after retrotransposition burst.

## 1. Introduction

Transposable elements (TEs) are DNA sequences capable of movement (transposition) and replication within the genome. These elements are present in the genomes of almost all eukaryotes [1]. In plants, TEs may comprise more than 90% of the genome, and this value often correlates with genome size [2]. All TEs are usually divided into two classes: class I, or retrotransposons, which replicate via an RNA intermediate according to the “copy-paste” principle, and class II, or DNA transposons, which move via “cut-paste” mechanisms [3,4,5]. Retrotransposons, particularly LTR retrotransposons, are usually the most prevalent TEs in plant genomes [6]. The structure and mechanism of movement of these elements make them unique modifiers of the genome, contributing to multiple scenarios of genome rearrangements, epigenetic changes, and transcriptome rewiring [7,8,9]. The activity of TEs in the genome is tightly regulated by multiple epigenetic and small-RNA-based mechanisms [10]. The mode of action of these mechanisms mainly depends on the TE location in the genome relative to genic sequences and chromosomal landmarks such as heterochromatin and centromeres [11]. Control of heterochromatic TEs is mainly mediated by the protein (DECREASE IN DNA METHYLATION 1) DDM1, whose remodeling action changes chromatin to make it accessible for METHYLTRANSFERASE 1 (MET1) and CHROMOMETHYLASE 2/3 (CMT2/3) [12,13]. In contrast, euchromatic elements are the target of the RNA-directed DNA methylation (RdDM) system, which, once established, operates on the principle of a self-reinforcing loop [14]. Due to the preference of some TEs for insertion into gene-rich regions, the RdDM system is balanced by specific proteins such as REPRESSOR OF SILENCING 1 (ROS1) and DEMETER (DME), which provide a compromise between TE silencing and transcription of nearby genes [11].

TE insertions are not randomly distributed in plant genomes. Some TE families tend to insert into pericentromeric regions (e.g., chromoviruses clade of *Ty3*/*Gypsy* LTR retrotransposons), while other TEs preferentially integrate into genic regions [15,16,17]. The latter group of TEs has received a lot of attention during the last decades and includes the well-characterized LTR retrotransposons *ATCOPIA78*/*ONSEN* and *ATCOPIA93*/*EVADE* of *Arabidopsis thaliana* [18,19]. Genic insertions of TEs often serve as raw material for evolution, leading to new regulatory repertoires and transcription modulation of the affected genes [6,20]. Prominent examples of gene expression enhancement include the blood-red color of orange flesh and red apple skin, where *Ty1*/*Copia* and *Ty3*/*Gypsy* elements act as cold-activated transcriptional regulators [21,22]. Another example of an LTR as a stress-inducible but naturally tethered promoter is the solo-LTR *ATCOPIA93*/*EVADE*, which mediates pathogen-activated expression of the *RECOGNITION OF PERONOSPORA PARASITICA 4* (*RPP4*) gene [23]. Insertions of LTR-RTEs near genes can also negatively affect their expression by locally altering the chromatin state and/or directing PTGS against a nearby gene [24,25]. RTEs can be integrated inside a gene, where they can also generate a variety of consequences on epigenetic and transcriptional levels [6,16,26,27,28]. The latter includes modification of splicing and polyadenylation signals as well as transcription start and termination sites [26,29,30]. TEs located within introns usually form heterochromatin, which is enriched in CHG and H3K9 methylation [31]. Alternative splicing in these regions is regulated via these repressive marks, and loss of intron heterochromatin leads to premature polyadenylation or cryptic transcription of associated genes [30,31,32]. Proper transcription and mRNA maturation of heterochromatin-containing genes depend on several chromatin regulators, including INCREASE IN BONSAI METHYLATION 2 (IBM2) and ASI1-IMMUNOPRECIPITATED PROTEIN 1 (AIPP1), which interact with intronic heterochromatin to promote distal polyadenylation [33].

More than 80% of intronic TEs are fragmented sequences of ancestral copies, and therefore the current picture of the TE insertion effect on transcription has been shaped by hundreds of years of evolution [31]. For example, the insertion of *ATCOPIA78*/*ONSEN* into the *FLC* Ag-0 gene supposedly occurred 246 years ago, which may be more than 400 generations from the moment of transposition [34]. Little is known about the epigenetic and transcriptomic changes that occur in a short time (a few generations) after the TE insertion takes place in a wild-type genetic background.

New insertions of *ONSEN* elements in *Arabidopsis thaliana* are still found in natural populations of *A. thaliana* [35]. Interestingly, both natural and experimentally induced *ONSEN* insertions are found predominantly within genes, making them especially valuable for studying the effects of transposition on gene structure and expression [16,28,35,36,37].

Moreover, *ONSENs* possess long terminal repeats (LTRs) that contain heat-responsive elements (HREs), which allow them to be activated by heat stress factors [38,39,40,41]. This property enables researchers to manipulate *ONSEN* transposition in a controlled manner and makes the *ONSEN* family very suitable for studying the short-term effects of TE insertions [18]. However, *ONSEN* activity is suppressed by the RdDM pathway, and controlled *ONSEN* transposition requires the suppression of RdDM. This has been achieved by using mutants in key components of RdDM (e.g., *nrpd1* mutants lacking the large subunit of PolIV) or by the application of chemical inhibitors of RdDM (e.g., alpha-amanitin inhibiting Polymerase II activity [42]). The latter approach is valuable as it allows the creation of novel insertions in wild-type genetic backgrounds. Thieme et al. developed and used this method for controlled *ONSEN* transposition without introducing mutations, resulting in the creation of high-copy *ONSEN* lines (hcLines) of *A. thaliana* [28,42]. Other research groups are also generating similar lines to study the patterns and consequences of transposition using advanced technologies [36,37]. Notably, Oxford Nanopore Technology (ONT) sequencing now allows detailed analysis of TE-gene transcripts and the methylation patterns of specific TE copies, providing new insights into the impact of transposition [26,37,43,44].

In this study, we obtained a collection of wild-type *A. thaliana* (Col-0) plants possessing between 2 and 32 novel *ONSEN* insertions, introduced by the combination of heat stress and drug treatment. To determine methylation and transcriptome changes caused by novel *ONSEN* copies, we utilized whole-genome and cDNA nanopore sequencing. Our results showed that full-length *ONSEN* element insertions in genes most frequently become introns during transcription, reducing the appearance of TE-gene transcripts. Less frequently, the novel insertions provided alternative transcription start and termination sites, leading to the generation of distinct gene-TE transcriptional isoform variants. Genome-wide methylation analysis of these plants revealed strong and precise methylation of novel *ONSEN* copies, with highly unequal distribution of CHH and CG methylation along the *ONSEN* sequences. The major factor influencing *ONSEN* insertion methylation was their location (euchromatin or pericentromeric heterochromatin) in the genome. In conclusion, our results demonstrate that DNA methylation and alternative splicing are effective control mechanisms safeguarding the genome and transcriptome integrity in plants.

## 2. Results

### 2.1. Obtaining A. thaliana Plants with an Increased Number of ONSEN Insertions

Using a modified TEgenesis protocol [42], we obtained M1 seeds that originated from nine heat-stressed M0 plants grown in plant growth (Z&A) medium, containing zebularine (a DNA methylation inhibitor) and alpha-amanitin (an inhibitor of Polymerase II). The M1 seeds showed a highly variable germination rate (from 10 to 80%). By performing qPCR, we detected an increased number of *ONSEN* copies both between and within the progeny of individual M0 plants (Figure 1A). According to the qPCR results, the progenies of M0 plant #12 and plant #13 had the highest number of *ONSEN* copies (from 5 to 25, Figure 1A) among the analyzed M1 plants. The number of new insertions did not correlate with the germination or survival rate of M1 plants. To verify the results of qPCR, we carried out transposon display analysis, which also showed that the obtained M1 plants possessed additional copies of *ONSEN*. In concordance with qPCR results, the M1 plants of plant #12 and #13 had the highest number of novel *ONSEN* insertions (Figure 1B).

Thus, we successfully obtained a collection of *A. thaliana* plants with novel *ONSEN* insertions in the genome.

### 2.2. Whole-Genome Analysis of Novel ONSEN Insertions

To study the effect of new *ONSEN* insertions, we selected one M2 plant from the progeny of each individual M1 plant with an increased number of *ONSEN* elements as indicated by the qPCR and transposon display results. We performed whole-genome nanopore sequencing of six M2 plants and obtained between 279,922 and 2,406,642 reads with the N50 ranging from 1444 to 12,405 bp. The corresponding genome coverage values ranged from 13 to 70×. To identify the genomic location of novel *ONSEN* insertions, we used a nanotei pipeline followed by manual curation of the detected TE insertions (TEIs) [45]. In total, we identified 56 novel *ONSEN* insertions among six M2 plants. The number of insertions varied from 2 to 32 per plant (Table 1 and S1). The highest number of novel insertions was detected for the M2 plant #12-1 originating from M0 plant #12. For five novel *ONSEN* insertions, we performed PCR validation (Appendix A). Although we did not perform zygosity assessment in previous generations, slightly less than half (27) of the inherited insertions were heterozygous. The intersection of the TEI locations and gene annotation showed that 51 TEIs (91.1%) were located in genes (Appendix A). These observations are consistent with previous data obtained by us and other authors, which showed that *Arabidopsis Ty1*/*Copia* elements, in particular *ONSEN*, prefer to insert into genes [16,28,36,37].

Next, we tested how the genomic distribution of heritable *ONSEN* insertions differs from that of somatic insertions identified by Cas9-directed sequencing (CANS) in an independent set of plants from the same experiment (Appendix A) [37]. We found a high correlation (two-sample Kolmogorov–Smirnov test, *p*-value = 0.2), demonstrating that the distribution of the detected heritable *ONSEN* insertions is similar to that observed for somatic *ONSEN* insertions, which also tended to be euchromatic (Figure 2).

Taking into account the assembly gaps, it is generally accepted that the *ONSEN* family in *A. thaliana* consists of eight full-length copies, three of which (*ONSEN1*/AT1TE12295, *ONSEN2*/AT3TE92522, and *ONSEN3*/AT5TE15240) are the youngest and share completely identical LTR sequences [38]. Using unique single-nucleotide polymorphisms (SNPs) in the RT domain of the original *ONSENs*, we were able to identify the parental copies for 42 of the 56 new insertions (Figure 3, Appendix A). According to our data, 85% of all insertions were produced by four *ONSEN* elements: *ONSEN1* (12 insertions), *ONSEN5* (11 insertions), *ONSEN3* (8 insertions), and *ONSEN2* (5 insertions). The rest of the TEIs possess *ONSEN* copies with SNPs shared by *ONSEN1* and other copies (*ONSEN3*, *ONSEN5*, or *ONSEN6*), making it impossible to establish their exact origin. Whether these insertions are the result of recombination between RNAs or cDNAs of two elements remains to be determined [46]. Nevertheless, these observations are consistent with previous studies, where *ONSEN1* and *ONSEN5* were the most efficient producers of new insertions [28,36]. These results also correlate with the amount of extrachromosomal circular DNA (eccDNA) these *ONSEN* elements form upon activation [37,38].

By comparing the gene lists of the inherited (51 genic insertions in this study, Appendix A) and somatic *ONSEN* insertions (432 genic insertions for 415 unique genes, Appendix A), we found 13 common genes. This value is significantly higher than expected by chance (Fisher’s exact test *p*-value = 8.20 × 10^−14^), providing additional evidence for the existence of insertion “hot spots” for *ONSEN*.

It was previously shown that *ONSEN* elements preferentially insert into genomic loci with a chromatin state enriched in H3K27me3 and the histone variant H2A.Z [16,28,36,37]. Thus, it can be suggested that the large number of common genes with heritable and somatic insertions may be explained by the similarity of the chromatin state in their bodies. To confirm this assumption, we performed enrichment analysis of the *ONSEN* insertion sites using information about chromatin states of individual *Arabidopsis* genes (Figure 4, Appendix A). We found significant enrichment of the inherited insertion sites (18 of 56, binomial test *p*-value = 7.483 × 10^−15^) and somatic insertions sites (123 of 517, binomial test *p*-value = 3.991 × 10^−75^) with chromatin state 11 from the Plant Chromatin State Database [47]. This state is characterized by the enrichment of H3K27me3, H2A.Z, and H3K4me2 and is most frequent in exons/coding regions of genes. To further validate our observations, we also performed an analysis of 237 *ONSEN* insertion sites from hcLines [28]. We found that 42 sites in hcLines belonged to chromatin state 12 (binomial test *p*-value = 1.669 × 10^−21^) enriched in H3K27me3 and H2A.Z, which is equivalent to the previously described chromatin state 5 [28,48]. Intriguingly, all preferred sites for *ONSEN* insertions are not characterized by the predominance of H2A.Z histone alone but are always represented by its co-presence with other epigenetic marks.

Next, we carried out gene ontology (GO) enrichment and functional analysis of the genes carrying inherited insertions. We found that four genes harboring *ONSEN* insertions (AT4G00190, AT5G20860, AT5G51500, and AT4G00730) are related to the cell wall modification pathway (GO:0042545, FDR = 3.5 × 10^−12^), while the remaining pathways were enriched by even fewer genes and had a lower FDR (Appendix A). Interestingly, despite the lack of enrichment dominance in specific pathways, some of the genes disrupted by *ONSEN* are those that are extremely important for growth and development, such as *SUPPRESSOR OF OVEREXPRESSION OF CONSTANS1* (*SOC1*), *RELATED TO ABI3/VP1 2* (*RAV2*), and *ASYMMETRIC LEAVES 2* (*ASY2*) (Appendix A).

Thus, we showed that *ONSEN* insertions are non-randomly distributed along chromosomes and preferentially located in chromatin enriched by the combination of H2A.Z and H3K27me3 and H3K4me2.

### 2.3. Transcription Changes Induced by ONSEN Insertions

To understand whether *ONSEN* insertion leads to changes in transcription, we performed nanopore sequencing of cDNA from leaf tissues of control and M2 plants (#7-5, #12-1, and #14-2) carrying *ONSEN* insertions. In total, we obtained between 1,026,345 and 1,720,022 ONT reads per sample. We first analyzed the transcription of the original eight *ONSEN* elements. For *ONSEN* elements located in proximity to annotated genes (*ONSEN2* and *ONSEN3*) or in the intron of a gene (*ONSEN1*), we detected chimeric transcripts containing parts of *ONSEN* sequences (Figure 5). Transcripts of the genes AT1G11270 and AT3G61320 possessed LTR sequences of *ONSEN1* and *ONSEN2* elements, respectively (Figure 5A,B). Gene AT5G13200 had two major isoforms, one of which included part of the 3’ LTR of the *ONSEN3* element. The 3’ end of the second isoform of AT5G13200 corresponds to part of the GAG-PROT sequence of *ONSEN3* (Figure 5C). Thus, for these genes, the nearby *ONSEN* elements serve as a source of transcription termination sites.

Next, we evaluated the systematic effect of novel *ONSEN* insertions on the transcription of the genes carrying the insertions. A significant number (40.7%) of genes with *ONSEN* insertions were not transcribed in either Col-0 or M2 leaves. We analyzed transcriptional changes for the TEIs whose expression was detected by sequencing (Appendix A). These TEIs were located in the exons as well as introns of the genes and resulted in distinct transcriptional changes. Among eight homozygote TEIs, three intronic TEIs underwent intronization of full-length *ONSEN* insertions (example in Figure 6A). Notably, the full-length *ONSEN* intronization has also been detected for the exonic insertion in the gene AT5G32415 indicating that LTRs of *ONSEN* elements may consist of splicing sites (Figure 6B). For seven TEIs, we detected partial exonization of the inserted *ONSEN* sequences resulting from alternative transcription termination (ATT) or alternative transcription start (ATS) sites. In these cases, short transcripts with disrupted open reading frames were observed (examples in Figure 6C,D). The exonization events were verified by PCR analysis (Appendix A). Notable examples of diverse *ONSEN* exonization events are the TEIs in the *SOC1* (AT2G45660) and *RAV2* (AT1G68840) genes of #12-1 plant. The *ONSEN* insertion in the *RAV2* gene caused significant changes in the transcription of this gene by creating ATT sites that led to short transcript generation (Figure 6C). The *ONSEN* derived isoform of *RAV2* is the major one based on the estimated isoform specific expression level (Figure 7). The transcripts of the *SOC1* gene with an *ONSEN* insertion demonstrated different scenarios of *ONSEN* sequence exonization resulting from ATT and ATS events. The orientation of TEI was in the opposite direction to that of the *SOC1* gene, and the chimeric isoforms were formed with parts of the antisense *ONSEN* strand (Figure 6D).

ATT events occurred when *ONSEN* RT-RH, GAG-PR, or R regions of LTRs served as transcription termination sites. The ATS events resulted from transcription start sites occurring within the antisense strand of the GAG-RT region of *ONSEN* (Appendix A). While ATSs within *ONSEN* LTRs and GAG-PR sequences have also been described previously [26,49]; other transcription patterns had not been observed previously.

We also predicted the open reading frames (ORFs) of TE-gene transcripts to analyze encoded proteins. Among the predicted ORFs for *RAV2* TE-gene transcripts, we identified truncated protein sequences with a portion of the translated long terminal repeat (LTR) region present at the C-terminus, resulting in chimeric proteins (Appendix A). For *ANL2*, the predicted ORFs were terminated either before or within the *ONSEN* sequence, leading to truncated or chimeric proteins as well. Additionally, in both cases, we detected small ORFs corresponding to regions of individual *ONSEN* domains.

The results demonstrate that novel *ONSEN* insertions generally do not significantly alter the composition of major isoforms. Moreover, splicing resulting in *ONSEN* intronization may serve as an effective host defense mechanism against new TE insertions, thereby ensuring transcriptome stability.

### 2.4. DNA Methylation of ONSEN Elements and Their Insertions

To investigate the variability of methylation patterns of novel *ONSEN* insertions, we analyzed the ONT sequencing data of six M2 plants (plants #12-1, #1-1, #3-6, #7-5, #13-1, and #14-2) and Col-0 as the wild-type.

#### 2.4.1. Methylation in Original Full-Length ONSEN Elements

To evaluate methylation levels of original full-length *ONSEN* copies, we analyzed mean binned methylation across samples. We found that CG and CHG methylation levels were higher in the *ONSEN* body and lower at the edges, while CHH methylation displayed the opposite pattern, with higher levels at the TE boundaries (Figure 8A, Appendix A). While the bodies of original *ONSEN* copies were overall hypermethylated, the flanking regions were often hypomethylated in all three contexts except for some methylation islands (Figure 8A, Appendix A). However, particular deviations from the general methylation patterns were found in *ONSENs* AT3TE54550, AT1TE59755, and AT3TE89830, where the CHH level did not show a characteristic increase in TE boundaries, remaining the same as in the TE body or even TE flanks (Figure 8A). This may be associated with the chromosomal position of these *ONSENs*. Indeed, AT3TE54550 and AT1TE59755 are located in the (peri)centromeric regions of Chr3 and Chr1, respectively. We compared the methylation profile of these *ONSEN* elements with the methylation of the pericentromeric LTR retrotransposon *EVD5* (AT5TE20395/*ATCOPIA93*) and found very similar patterns (Figure 8A). We next checked the connection between the methylation of *ONSEN* elements and the number of insertions they generated. In our data, we do not observe a clear connection between the original *ONSEN* copy methylation pattern and the number of insertions it produced. However, for two *ONSENs* with no insertions in any sample (AT1TE24850 and AT1TE71045), the methylation shows the lowest standard deviation, which reflects the stability of the pattern in different cells (Figure 8A,B). This suggests that the generation of new copies (TEIs) somehow increases variability in the original copy methylation level. Indeed, the more insertions *ONSEN* generates, the more variable its methylation becomes (Figure 8C,D), even though mean *ONSEN* methylation does not correlate with the number of insertions (Appendix A).

#### 2.4.2. Methylation in Novel ONSEN Insertions

To analyze methylation patterns of novel *ONSEN* insertions, we modified the TAIR10.1 reference genome and annotation by incorporating detected *ONSEN* sequences, creating “pseudoreferences”. This approach allowed us to map ONT sequencing data to these pseudoreferences and assess methylation calls within the TEI regions, including the insertions themselves (Figure 9).

Our analysis revealed that novel copies of *ONSEN* elements are hypermethylated in all sequence contexts (CG, CHG, and CHH) (Figure 10A, Appendix A). The inserted *ONSENs* had the methylation patterns that were consistent with the methylation of the original *ONSEN* elements. CG methylation was uniformly high, except at the boundaries, where it sharply decreased to zero. CHG methylation followed a similar trend but at lower mean levels. In contrast, CHH methylation exhibited an inverse pattern, with low to moderate levels in the TEI center and increasing levels toward the LTR and TEI boundaries.

We next asked how TEI location influences the pattern of DNA methylation. We found that the methylation pattern of *ONSEN* elements in the TEIs differs between the pericentromere and chromosome arm. While the CHG/CHH methylation level was close to zero over the *ONSEN* elements in pericentromeric TEIs, the TEIs from the chromosome arms exhibited clear LTR peaks of CHG/CHH methylation, indicating RdDM targeting (Figure 10B). Moreover, euchromatic TEIs generated by heterochromatic (pericentromeric) *ONSEN* elements (*ONSEN5* and *ONSEN8*) have clear RdDM methylation patterns that differ significantly from the original *ONSEN* elements (Figure 10A). Similarly, the pericentromeric TEIs generated by euchromatic *ONSEN* elements have high levels of CG methylation and very low levels of CHH/CHG methylation. These observations showed that the TE location rather than TE sequence determines which DNA methylation system is involved in the TE silencing.

We found that the methylation pattern of the inserted *ONSEN* copies vary more than the original ones and shows a lower mean methylation level, especially for the CG context. This observation probably reflects both the variability in the location of TEIs and the instability of TEI methylation. At the same time, similar levels of CG, CHG, and CHH methylation patterns (as for AT3TE92525 TEIs, see Figure 10A) could originate from the mechanism of initial TE methylation by the RdDM pathway methyltransferases *DRM1* and *DRM2*, which modify cytosines in all three contexts [11,50]. This suggests that, potentially, methylation of novel *ONSEN* copies is still at the stage of establishment or initiation rather than maintenance.

Previous studies have shown that RdDM-mediated methylation can spread beyond the TE boundaries and affect neighboring genes [51,52]. Indeed, in some of the TEIs analyzed, CHH methylation extends asymmetrically beyond the insertion boundaries forming continuous tracks, as evidenced by the difference in methylation levels between #12-1 TEI flanks and their corresponding regions in Col-0 (Figure 10C). Even though for most of the insertions these tracks do not last longer than 0.5–1 kb, there are some *ONSEN* insertions where the border methylation extends to at least 2 kb from the TE element (Appendix A). The direction of the CHH methylation spread cannot be explained solely by TEI or gene direction, which could result in sense–antisense transcription (Appendix A). Even though previous studies had also demonstrated the asymmetric spread of TE body methylation on the flanking regions [52], the mechanism of this asymmetry emergence remains a question to address.

## 3. Discussion

In this study, we generated a collection of *Arabidopsis* plants harboring novel *ONSEN* insertions and used this resource to investigate the DNA methylation and transcriptional changes induced by these events. Our results show that new insertions can influence splicing patterns in various ways; however, in most cases, they do not significantly alter the overall isoform composition. Notably, novel insertions are frequently intronized, suggesting that alternative splicing serves as an additional mechanism to mitigate the detrimental effects of TE insertions on gene transcription. Furthermore, we found that these new insertions are subject to significant DNA methylation, with the specific methylation pathway depending on the genomic location of the TE insertion.

### 3.1. Convergent ONSEN Insertions in Different Arabidopsis Collections

We found that the distribution of the detected *ONSEN* insertions did not differ significantly from the somatic *ONSEN* insertions detected by Cas9-mediated sequencing of independent M0s [37]. Moreover, we identified a high number of common genes selected for *ONSEN* integration in both cases. Given the preference of *ONSEN* for certain chromatin states [16,28], including those formed in response to heat stress [37], we can speculate that the intersection between somatic and heritable insertions is determined by similar growth conditions and the developmental stage when heat stress was applied. We also found that many genes with new inherited *ONSEN* insertions are not expressed in either Col-0 or our lines under non-stress conditions but are stress-inducible or expressed at certain developmental stages. This is consistent with a previous study of hcLines and suggests that preference for a certain chromatin state allows transposition to avoid constitutively expressed core genes, limiting the possibility of plant fitness reduction, which was discussed previously [27,28,35].

### 3.2. Genomic Location Dictates DNA Methylation Landscapes of Arabidopsis ONSEN Elements 

Studying DNA methylation changes induced by novel insertions in wild-type plants is challenging because of (1) the repetitive nature of the inserted TEs and (2) the complicated process of the triggering of TE transposition events and their inheritance to the subsequent generation. To overcome these limitations, we applied Oxford Nanopore sequencing to profile methylation of both original and novel *ONSEN* insertions in *Arabidopsis* plants that underwent chemical activation to induce *ONSEN* transposition [42]. Because of this, we were able to obtain a detailed genome-wide picture of DNA methylation distribution over original *ONSEN* elements and their insertions. We found that novel insertions have strong DNA methylation. Additionally, the pattern of DNA methylation is substantially different between euchromatic and heterochromatic insertions. While euchromatic original *ONSENs* have high CHH methylation levels at the edges and reduced CG and CHG methylation in the body, pericentromeric *ONSEN5* and *ONSEN8* have high CG levels, with a low and uniform CHH distribution. Moreover, ‘euchromatic’ insertions of initially ‘heterochromatic’ *ONSEN* elements lead to a significant deviation in the DNA methylation pattern of the inserted elements compared to the original ones. These observations demonstrate that the genomic location of a TE determines how the TE will be methylated. This is consistent with the current model of methylation mechanisms, where CHH methylation of euchromatic TEs, specific to chromosome arms, is mainly mediated by DOMAINS REARRANGED METHYLASE 2 (DRM2) targeted via the RdDM pathway, whereas methylation of heterochromatic TEs is maintained by *CMT2* [11]. RdDM is thought to target LTR regions, resulting in a specific increase in CHH methylation at the edges of TEs [53].

### 3.3. Distinct Phases of DNA Methylation of Novel ONSEN Insertions

The methylation of novel TE insertions is a dynamic process and includes initiation, establishment, and maintenance phases [54]. Clearly, the detected *ONSEN* insertions have already passed the first phase as they are already methylated. Can we observe the TEIs that are in distinct phases in our *Arabidopsis* plants? Initiation of DNA methylation of novel TE copies is triggered by the non-canonical PolII-RdDM pathway. This includes PolII-mediated TE expression, 21–24 siRNA generation by one of the DCL proteins, and an AGO4-clade protein-mediated siRNA base pairing with nascent PolII transcripts in the nucleus followed by PolV recruitment and DNA methylation by *DRM1/2* [55]. This results in PolII-RdDM initiation, and the hallmark of this stage is DNA methylation in all three contexts over the entire TE body [56]. The only *ONSEN* insertions that exhibit a similar DNA methylation pattern in our assay originate from *ONSEN2* (AT3TE92525). However, the CHH methylation peaks at the LTRs also suggest that the next phase has already begun. The establishment phase of DNA methylation does not depend on PolII transcription and relies on the PolIV-RdDM pathway [57]. The euchromatic TEs at this phase of DNA methylation can be distinguished by a non-random distribution of cytosine methylation over the TE body, with LTR sequences mostly methylated at the CHH context and minor CG methylation, suggesting targeting by the PolIV-RdDM pathway [55]. However, the inter-LTR TE sequences are mostly methylated in the CG context which may indicate the maintenance of DNA methylation via MET1 action. Based on this, we conclude that DNA methylation of almost all of the detected euchromatic *ONSEN* copies is under the double control of RdDM and *MET1*. Alternatively, novel *ONSEN* insertions can be methylated via identity-based DNA methylation mechanisms [56].

We noted that the most active original *ONSENs*, as well as their novel TEIs, have a high level of methylation standard deviation, most pronounced in the CG context. It is known that chemical treatment with zebularine leads to DNA hypomethylation in all contexts [58], which cannot be completely eliminated in the first generations [36,59]. In this case, some portion of the lost symmetric methylation in the TE body should be recovered de novo through the RdDM pathway [60]. Although PolII-derived siRNAs can act in trans to lead to de novo methylation of all TE family members, it is difficult to assess the impact of the sequence specificity of the siRNA pool for the sequence to be methylated de novo [61]. Thus, the high deviations of methylation observed in the most active TEs and their new copies across samples may result from stochastic targeting of the siRNA pool to maintain methylation of a sharply increased number of highly similar *ONSEN* sequences. This is well supported by how stable methylation is maintained in *ONSEN4*, *ONSEN7*, and *ONSEN8*, which have not undergone transposition in our plants.

### 3.4. Dual Defense: DNA Methylation and Alternative Splicing Rapidly Mitigate ONSEN Insertions in Arabidopsis

Transposon sequences are globally represented in the *Arabidopsis* transcriptome by thousands of chimeric TE-gene transcripts [26]. It is worth noting that the creation of new introns, as well as other transcriptomic changes driven by TE insertions, have contributed greatly to the phenotypic variation of other *Brassicaceae* species [62,63]. Using the potential of long-read nanopore sequencing of cDNA, we assessed transcriptome changes in sites of very recent *ONSEN* insertions at the isoform level. Surprisingly, for most of the sites, we observed only minor changes in transcription and isoform sequences introduced by *ONSEN* insertions. Among exonic and intronic insertions, the former most often lead to the generation of TE-gene transcripts. We found that such transcripts resulted from alternative transcription termination or alternative transcription start sites located in *ONSEN* sequences. This observation is in good concordance with the previous reports about *ONSEN*-mediated transcription induction found in natural *Arabidopsis* accessions [49]. Unexpectedly, we also observed full-length *ONSEN* intronization in the case of exonic insertion in gene AT4G34215. Although certain TEs can independently create cryptic introns [64], in the case of *ONSEN* this can be explained by the unique context of this insertion. Unlike most exonic insertions, *ONSEN* intronic insertions were more often completely excluded from gene transcripts by alternative splicing. To further investigate the consequences of this transcript diversity, we analyzed predicted ORFs in TE-gene transcripts from the *RAV2* and *ANL2* genes. This analysis suggests that the incorporation of TE sequences can result in truncated or chimeric proteins as well as to full-length or fragmented transposon-derived proteins, whose biological roles and regulatory mechanisms remain largely unexplored.

The described alternative splicing events may interconnect with the hypermethylation of novel *ONSEN* insertions through several mechanisms. First, transcription of genes with long heterochromatic introns is known to be targeted by IBM1/IBM2, which promote proper splicing of TE-containing introns, as demonstrated for *ONSEN* insertions in the *ABI5* gene of *Arabidopsis* [65]. Moreover, splicing of the *IBM1* gene itself is regulated via methylation of its TE-derived intron [66]. *IBM1*’s role in gene expression regulation is further highlighted by its impact on meiotic gene expression and fertility in *Arabidopsis* [67]. Second, hypermethylation may slow down Pol II progression through gene bodies, allowing spliceosomes more time to recognize suboptimal splice sites [68,69,70]. This delay may increase the inclusion of alternatively spliced exons. For example, H3K9me2-marked chromatin (associated with DNA methylation of TEs) correlates with slower Pol II and altered AS patterns in mice [71]. While the link between differentially methylated genes and differentially spliced genes has not previously been detected in plants [26], a mutation in the rice DNA methyltransferase gene *OsMet1-2* globally affected alternative splicing [26]. Finally, H3K9me2 histone modification has been associated with R-loop formation, leading to alternative polyadenylation and transcription termination in plants [72,73]. R-loops have also been shown to form during RdDM action resulting from Pol IV transcription [73] or *AGO4*-mediated siRNA interaction with DNA [74]. It is intriguing to test whether inhibition of R-loop formation will lead to transcription changes in the *ONSEN* insertion sites.

Altogether, our results show that, similar to DNA methylation, the recognition and intronization of *ONSEN* insertions are established rapidly after their origin. This quick “transcriptome defense” system may be considered an additional layer of protection for genome and transcriptome integrity against transposon insertions.

## 4. Materials and Methods

### 4.1. Plant Material and Seeds Sterilization

Arabidopsis Col-0 seeds were surface-sterilized with 75% ethanol (for 2 min) and washed with 5% sodium hypochlorite (for 5 min). The seeds were then rinsed three times with sterile distilled water, resuspended in 0.1% agarose, and kept at 4 °C for 48 h for vernalization and to promote synchronous germination.

### 4.2. Transposon Activation and M1 Seed Production

For *ONSEN* activation in M0 plants, we applied a modified TEgenesis protocol originally proposed by Thieme et al. [42]. A suspension of vernalized seeds in agarose was placed on Petri plates containing Z&A medium: ½ MS (OJSC BioloT, Saint-Petersburg, Russia) supplemented with sterile-filtered 4.6 μg/mL α-amanitin (Sigma-Aldrich, St. Louis, MO, USA; CAS 23109-05-9) and 9 μg/mL zebularine (Sigma-Aldrich, St. Louis, MO, USA; CAS 3690-10-6). Plants were grown under standard long-day conditions (22 °C; 16/8 h day/night photoperiod). For heat stress treatment, seven-day-old Arabidopsis plants were transferred to 4 °C for 24 h and then to 37 °C for 24 h. After stress, plants were returned to standard growth conditions. After 4 days of recovery, the shoots were cut from the root system, rinsed with distilled water, and transferred into glass culture tubes containing Hg medium [75]. After in vitro seed propagation and maturation, the M1 seeds were collected.

### 4.3. M1 and M2 Plants Growth Conditions

All plant progenies were grown under standard long day conditions (22 °C; 16/8 h day/night photoperiod). M1 plants were grown on ½ MS (OJSC BioloT, Saint-Petersburg, Russia), and M2 seeds were harvested and were sown in a 10 × 10 cassette containing a soil–perlite mix (3:1).

### 4.4. DNA, RNA Isolation, and cDNA Synthesis

Plant leaves weighing approximately 200 mg were homogenized in liquid nitrogen using a mortar and pestle and then divided into equal portions for total DNA and RNA isolation. Total DNA was isolated using a modified CTAB protocol [76]. Briefly, 0.5 mL of CTAB1 buffer, preheated to 75 °C and containing 6% β-mercaptoethanol and 0.5% polyvinylpyrrolidone, was added to the frozen powder. The lysate was transferred to a 1.5 mL tube and incubated at 75 °C for 1 h. After cooling, an equal volume of chloroform was added, and the mixture was centrifuged. The upper aqueous phase was transferred to a new 1.5 mL tube containing two volumes of CTAB2 buffer. The resulting pellet was resuspended in 0.2 mL of 1 M NaCl, and DNA was precipitated with an equal volume of isopropanol, followed by centrifugation. The pellet was washed with 70% ethanol and resuspended in nuclease-free water. RNase treatment was performed, followed by isopropanol reprecipitation and ethanol washing. The final DNA pellet was resuspended in nuclease-free water for downstream analyses.

Total RNA was isolated using the ExtractRNA kit (Evrogen, Moscow, Russia) according to the manufacturer’s instructions. For cDNA synthesis, 500 ng of total RNA was reverse transcribed using oligo(dT)15 primers and the MINT cDNA kit (Evrogen, Moscow, Russia), following the manufacturer’s protocol. The resulting double-stranded cDNA was purified using 1.8 volumes of Agencourt AMPure XP beads (Beckman Coulter, Brea, CA, USA) in accordance with the manufacturer’s instructions.

### 4.5. qPCR, Transposon Display and PCR Analysis

Quantitative PCR (qPCR) was performed using 5 ng of total DNA per reaction with the BioMaster HS-qPCR (2×) master mix (Biolabmix, Novosibirsk, Russia) on a Tianlong Gentier 96E Real-Time PCR System. Primers targeting the *ONSEN* transposon RT-domain and the reference gene *ACTIN2* (AT3G18780) were employed [38]. Each sample was analyzed in triplicate, and cycle threshold (Ct) values were normalized to *ACTIN2* using the 2^−ΔΔCt^ method. Copy numbers were calculated relative to the Col-0 wild-type control, with data processed using LOCUS Intero software, version 1.0.282(1.0.95). The thermal profile included an initial denaturation at 95 °C for 3 min, followed by 40 cycles of 95 °C for 10 s and 58 °C for 30 s.

To analyze *ONSEN* transposon insertion sites, the transposon display (TD) method was used with modifications [77]. Briefly, 200 ng of total genomic DNA was digested with the DraI restriction enzyme (SibEnzyme, Novosibirsk, Russia). The digested DNA was purified using 1.8× volume of Ampure XP beads according to the manufacturer’s instructions and eluted in 30 μL of nuclease-free water. Genome Walker adaptor oligonucleotides (GW adaptor F/R) were synthesized and ligated to the digested, purified DNA fragments, followed by a 1:20 dilution with nuclease-free water. Ligated and diluted DNA (1 uL) was amplified using the BioMaster LR HS-PCR (2×) master mix (Biolabmix, Novosibirsk, Russia), a primer specific to the adaptor (AP1), and an *ONSEN*-specific primer (ONSEN_312 or ONSEN_3’LTR) targeting the LTR region [27,77]. Thermal cycling conditions included an initial denaturation at 95 °C for 5 min, followed by 33 cycles of 95 °C for 30 s, 58 °C for 30 s, and 72 °C for 1 min, with a final elongation step. Amplicons were resolved on 1.5% agarose through gel electrophoresis and visualized via ethidium bromide staining.

To validate novel transposon insertions and alternative splicing events, PCR amplification was performed using 5 ng of total genomic DNA or 1 ng of cDNA, respectively. Reactions were carried out with BioMaster HS-PCR (2×) master mix (Biolabmix, Novosibirsk, Russia), employing transposon-specific primers paired with flanking genomic primers to detect insertion junctions. For alternative splicing analysis, primers targeting constitutive exons and *ONSEN* sequences were designed to distinguish splicing isoforms. Thermal cycling included an initial denaturation at 95 °C for 5 min, followed by 35 cycles of 95 °C for 30 s, 58 °C for 30 s, and 72 °C for 1 min, with a final extension at 72 °C for 5 min. Amplicons were resolved on 2% agarose gels and visualized via ethidium bromide staining. Gel images were acquired using a Gel Doc XR+ Gel Documentation System (Bio-Rad, Hercules, CA, USA) equipped with trans-UV and epi-white illumination sources and a high-resolution CCD camera. Image capture and analysis were performed using Image Lab software, version 6.1, following automated protocols to ensure reproducibility and optimal image quality. Images were exported in TIFF format and post-processed in Adobe Photoshop (Adobe Inc., San Jose, CA, USA) with only linear global adjustments (cropping, brightness, and contrast) applied to the entire image. No non-linear or selective editing was performed.

All primers used in this study are listed in Appendix A.

### 4.6. Nanopore Sequencing

For nanopore sequencing, a library was prepared using Native Barcoding Kit 96 V14 (SQK-NBD114.96, Oxford Nanopore Technologies, Oxford, UK). Sequencing was performed on a Promethion P2 Solo equipped with a PromethION R10.4.1 flow cell.

### 4.7. Genome Assembly

In the current study, TAIR10.1 (RefSeq accession: GCF_000001735.4) genome assembly and Araport11 genome annotation was used [78].

### 4.8. Novel Insertions Discovery, Assembly, and Pseudoreference Generation

Raw ONT R10 pod5 data were basecalled with Dorado v0.7.2 (https://github.com/nanoporetech/dorado (accessed on 16 April 2025)) with the model dna_r10.4.1_e8.2_400bps_hac@v5.0.0 and the --emit-fastq option enabled for downstream analysis. Output fastq data were trimmed with default Dorado parameters, aligned using minimap2 v2.17 with the following options: “-ax map-ont options”, sorted, and indexed with samtools v.1.9 [79,80]. Obtained BAM files were visualized using JBrowse2 v1.6.5 [81]. For insertion discovery, the Nanotei pipeline was used with the following parameters: --fpv --bed --minpvalue 0.05 -ovt 0.3 [45]. Obtained insertions were filtered by number of supporting reads (min. 4) and TEI region coverage (IsTEIregIsOK, TRUE). All detected insertions went through visual inspection with JBrowse2 to validate insertion events, correct false negative Nanotei results, and refine TSD boundaries. For the assembly of the insertions in each barcode, all aligned reads within insertion boundaries were extracted and processed as follows: (1) the clipped (S flag in CIGAR string) or insertion (I flag in CIGAR string) part of the aligned read was extracted using pysam v.0.16.0.1 (https://github.com/pysam-developers/pysam (accessed on 16 April 2025)) and re *python* packages; (2) obtained sequences were aligned using mafft v.7.453 [82]. For further analysis these sequences were incorporated into the reference genome (TAIR10.1) with duplicated TSDs, generating a modified reference (‘pseudoreference’) for each barcode. Positions in the reference annotation (Araport11) were shifted accordingly.

### 4.9. Novel Insertions Annotation

Assembled insertions were annotated using DANTE v.0.1.9 with the default parameters and LTRharvest (GenomeTools v.1.6.1) [83,84].

### 4.10. Distribution of Novel Insertion Sites in the Arabidopsis Genome and Chromatin States

Coordinates of novel *ONSEN* insertion sites (this study, Appendix A), binned (20 kb) somatic *ONSEN* insertion sites (Appendix A), and *Arabidopsis* centromeric, pericentromeric heterochromatin, and euchromatic arm regions (Appendix A) were used to visualize the chromosome distribution of TAIR10.1 genome using the ChromoMap *R* package [37,85,86].

For chromatin state enrichment analysis, coordinates of novel *ONSEN* insertion sites (this study, Appendix A), somatic *ONSEN* insertion sites (Appendix A), *ONSEN* insertion sites from hcLines (Appendix A), and a set of 100 random sites per chromosome were intersected with chromatin states from the Plant Chromatin State Database [28,37,47]. The obtained count data were used for visualization.

### 4.11. Gene Ontology Analysis of Genes Harboring Novel ONSEN Insertions

Gene ontology (GO) enrichment analysis for genes with inherited *ONSEN* insertions was conducted using ShinyGO v0.82, focusing on biological processes from the GO database [87]. The analysis utilized the hypergeometric test to calculate enrichment *p*-values, followed by Benjamini–Hochberg false discovery rate (FDR) correction for multiple testing. Significant enrichment was determined using an FDR cutoff of 0.05, with results ranked by fold enrichment and statistical significance (Appendix A).

### 4.12. Determination of TEI Parental ONSEN Copies

Parental *ONSEN* copy for each TEI was identified using reverse transcriptase (RT) as the most conserved DNA sequence within TE. In brief, original *ONSEN* copies were annotated using DANTE, their RT sequences were aligned using mafft, and individual RTs derived from each TEI were aligned to obtained alignment (Appendix A). Belongingness of each TEI to one or several original *ONSEN* copies was determined based on the sum of presence/absence of 16 SNPs (Appendix A). Alignment was visualized using Unipro UGENE v.48.1 [88] for manual SNPs curation.

### 4.13. Methylation Calling

For R10 ONT data, methylation calling was performed with Dorado (v0.7.2) basecaller hac, 5mC_5hmC models (dna_r10.4.1_e8.2_400bps_hac@v5.0.0, dna_r10.4.1_e8.2_400bps_hac@v5.0.0_5mC_5hmC@v1) and –reference option to align each sample to its pseudoreference. Additionally, Col-0 samples were realigned to other samples’ pseudoreferences with the dorado aligner to further visualize methylation levels at insertion flanking regions.

Non-primary and supplementary alignments were filtered out with samtools view -F 2308. Consensus methylation calls were produced at each cytosine with modkit v0.4.4 (https://github.com/nanoporetech/modkit (accessed on 16 April 2025)) with the following options: “pileup–combine-mods--filter-threshold C:0.75”. Calls at positions with at least three reads were retained, and the remaining calls were split by cytosine context (CG, CHG, and CHH) using modkit motif-bed and bedtools v2.27.1 intersect [89]. For further analysis and visualization, cytosines with valid coverage below 0.05 or above 0.95 quantile were filtered out as outliers.

### 4.14. Methylation Analysis

To assess methylation levels in original and novel (TEIs) *ONSEN* copies in each sample, positions were taken from original and modified Araport11 annotations, respectively. The start of AT1TE24850 (*ONSEN7*) was shifted 2691 bp right to exclude a strongly hypomethylated region outside the annotated transcript. Methylation levels were combined between replicates of each sample and methylation context in 50 bp bins for flanking regions or 100 bins for TE bodies; then, the average methylation level was obtained for each bin. Additionally, to demonstrate the methylation difference in TEI flanking regions, the methylation in the Col-0 sample at the positions corresponding to TEI flanking regions in other samples was subtracted from TEI flanking methylation bin-wise.

For methylation heatmaps, binned methylation data were plotted separately for TE flanks and TE bodies.

For metaplots, the binned methylation levels in 2 kb flanking regions were combined with gene body data from multiple samples. *ONSEN* and TEI data were processed separately, with TEIs being matched to their parental *ONSENs* based on origin information. Additionally, TEIs were matched to their chromosomal position or other denoted features. Methylation levels were averaged across samples and smoothed using a 10-bin rolling mean. Standard deviation was calculated to represent data variability. In the resulting plots, mean methylation was displayed as solid lines for *ONSENs* and dashed lines for TEIs, with shaded regions indicating ±1 standard deviation. All plots were generated in Python (version 3.8.10) using matplotlib (except for methylartist methylation plot), with genomic positions annotated as −2 kb, start, end, and +2 kb relative to element boundaries.

### 4.15. Chimeric Gene-TE Transcripts Detection

Raw ONT RNAseq reads were trimmed using Porechop v.0.2.4 (https://github.com/rrwick/Porechop (accessed on 16 April 2025)) (parameters: “--require_two_barcodes”). Then, trimmed reads were mapped to the reference genome (individual pseudoreference) using minimap2 with the following parameters: “-a -x splice”, obtained SAM files were converted to BAM file, filtered (only primary alignments were taken for further analysis), sorted, and indexed using samtools. Transcripts were collapsed using StringTie2 v.2.2.1 [90] with “-R” parameter to obtain more chimeric transcripts [26]. Filtered and sorted BAM files as well as StringTie2 annotations were visualized in JBrowse2 v.1.6.5 [81] for manual data curation. After manual curation of insertion sites, they were characterized using Reddy et al. classification [91].

### 4.16. Analysis of Differential Isoform and Gene Expression

For differential gene expression analysis of gene–TE fusions, reads from two technical repetitions were counted using featureCounts v.2.0.0 [92] and modified Araprot11 GTF file (options: “-L--primary”). Counts per gene were normalized using TPM units with bioinfokit v.2.1.4 *python* package (http://doi.org/10.5281/zenodo.3698145 (accessed on 16 April 2025)). Differential gene expression analysis was performed using DESeq2 v.1.46.0 *R* package [93].

For differential isoform expression analysis, reads, corresponding to TEIs, were extracted using samtools view with the “-b-L” options and counted using samtools view with the “-c” option. Reads corresponding to a “normal” isoform (without any intersection with TEIs) were calculated as difference between all gene counts and TEI-corresponded counts. Counts were normalized using RPKM units with the bioinfokit *python* package.

Open reading frames (ORFs) were predicted using the NCBI ORFfinder tool (https://www.ncbi.nlm.nih.gov/orffinder/ (accessed on 16 April 2025)) to identify potential protein-coding regions within TE-gene transcripts. The analysis was performed using the standard genetic code and default parameters.

### 4.17. Visualization

For visualization, ggplot2 v.3.5.1 *R* package was used [94]. Read alignment visualization was performed in JBrowse2. Methylation heatmaps were plotted using matplotlib v.3.4.3 and seaborn v.0.11.2 *python* packages. For locus-specific per-read methylation calls’ visualization, methylartist v.1.3.1 [95] was used, using a region plot with parameters “-n C--motifsize 1--min_window_calls 0-m m”.

### 4.18. Data and Code Availability

Code for insertion reconstruction and pseudoreference generation as well as custom python code for Methylartist plots generation can be found at GitHub (https://github.com/shaperones/At_paper_Feb_2025 (accessed on 16 April 2025)).

## 5. Conclusions

Our study demonstrates that novel *ONSEN* insertions in *Arabidopsis* are rapidly targeted by DNA methylation and alternative splicing mechanisms, which together serve to minimize their impact on gene expression and transcriptome integrity. While these insertions can generate new splicing patterns and chimeric transcripts, most do not substantially alter overall isoform composition, and their effects are shaped by the genomic and epigenetic context of integration.

## Figures and Tables

**Figure 1 ijms-26-04816-f001:**
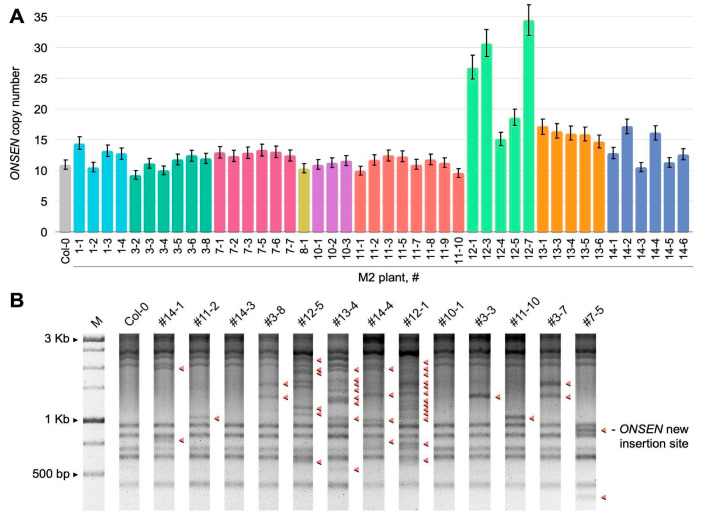
(**A**) qPCR evaluation of copy number variation of *ONSEN* elements in M2 plants. The colors of the columns represent the following genotypes: gray for the wild-type plant (Col-0), and for the progeny of M1 plants: light blue (#1), cyan (#3), pink (#7), yellow (#8), violet (#10), red (#11), light green (#12), orange (#13), and blue (#14); (**B**) transposon display detection of *ONSEN* insertions in M1 plants. Red arrows indicate additional copies of *ONSEN*.

**Figure 2 ijms-26-04816-f002:**
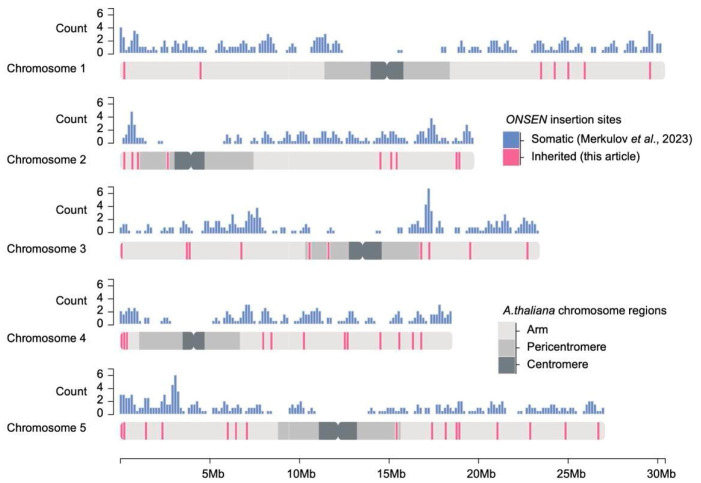
Chromosomal distribution of inherited (pink bands) and somatic (blue bars) *ONSEN* insertions. Somatic insertions were counted in 20 kb windows (see Section 4). Chromosome regions are indicated by shades of grey: light grey for the chromosome arm, medium grey for the pericentromeric region, and dark grey for the centromere [37].

**Figure 3 ijms-26-04816-f003:**
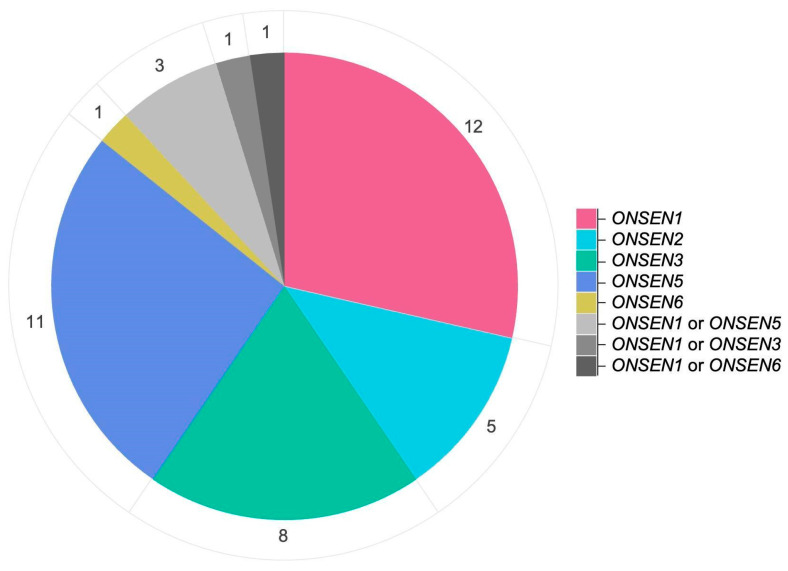
Number of new insertions in M1 plants generated by each of the eight *ONSEN* family members.

**Figure 4 ijms-26-04816-f004:**
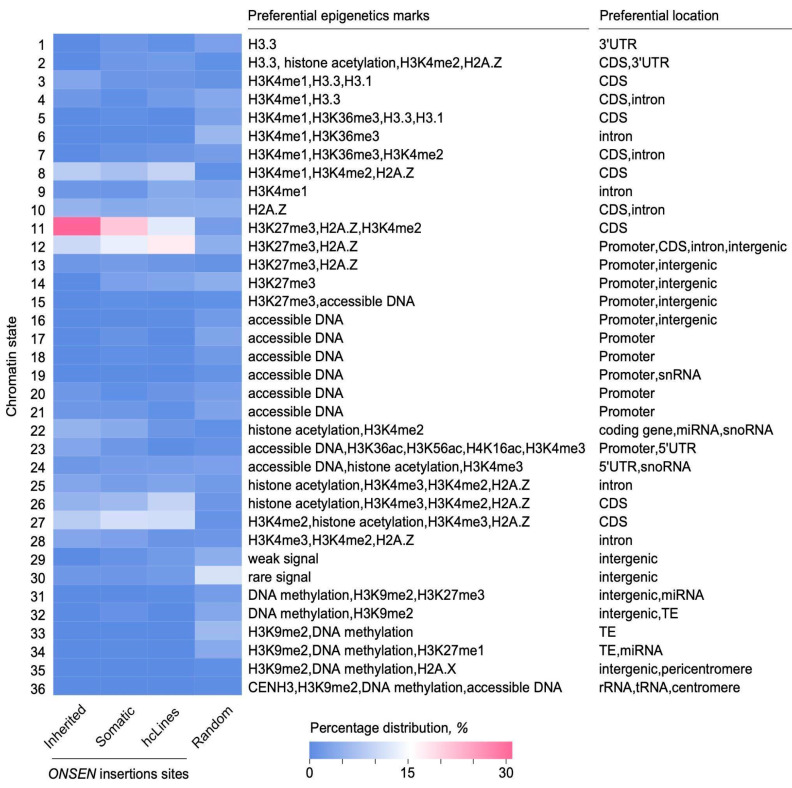
Heat map of the distribution of *ONSEN* insertions among chromatin states from the Plant Chromatin State Database [47]. Each row of the heat map corresponds to one of the 36 chromatin states, and a column corresponds to a set of insertion sites: heritable insertions (present study), somatic, hcLines, and random (a set of 100 random sites from each TAIR10.1 chromosome) [28,37]. The heatmap colors represent the percentage of overlapping chromatin state regions out of the total number of regions in each set. The characteristics of each chromatin state are presented in the table to the right of the heat map.

**Figure 5 ijms-26-04816-f005:**
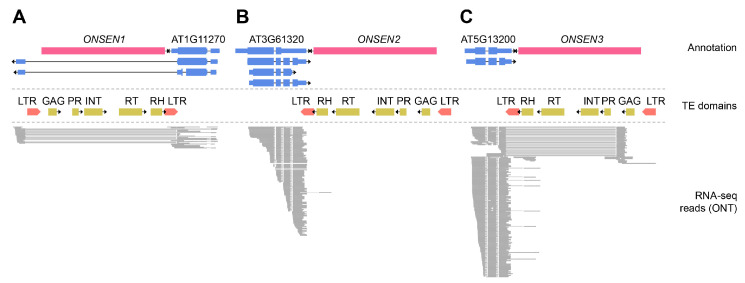
Transcription pattern of the original *ONSEN* elements, (**A**) *ONSEN1*, (**B**) *ONSEN2,* and (**C**) *ONSEN3*. The pink and blue blocks of ‘Annotation’ track indicate *ONSEN* elements and exons of the annotated genes. ‘TE domains’ track shows the position of LTRs and protein coding parts of *ONSEN* elements. The ONT cDNA reads aligned to the genome are shown in the “RNA-seq reads” track. Abbreviations of TE domains: LTR—long terminal repeat; GAG—group-specific antigen; PR—protease; INT—integrase; RT—reverse transcriptase; RH—RNase H. Arrows indicate the direction of transcription (5′ → 3′) for both genes and transposable elements (TEs), corresponding to the orientation in which the feature is transcribed on the DNA strand.

**Figure 6 ijms-26-04816-f006:**
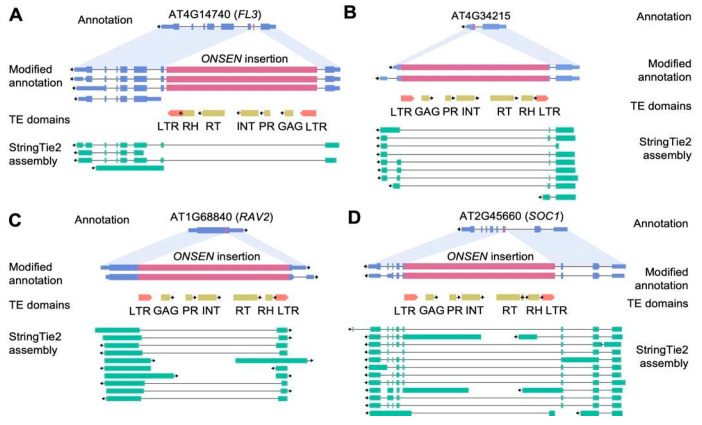
Examples of TEI (**A**) intronization of *ONSEN* insertion within existing intron, (**B**) intronization of ONSEN insertion in exon, (**C**) alternative transcription termination (ATT) event, and (**D**) alternative transcription start and alternative transcription termination (ATS and ATT) events. The pink and blue blocks indicate *ONSEN* elements and exons of the annotated genes. ‘TE domains’ track shows the position of LTRs and protein coding parts of *ONSEN* elements. The transcripts assembled by StringTie2 from cDNA ONT reads of the #12-1 are colored in cyan. Abbreviations of TE domains: LTR—long terminal repeat; GAG—group-specific antigen; PR—protease; INT—integrase; RT—reverse transcriptase; RH—RNase H. Arrows indicate the direction of transcription (5′ → 3′) for both genes and transposable elements (TEs), corresponding to the orientation in which the feature is transcribed on the DNA strand.

**Figure 7 ijms-26-04816-f007:**
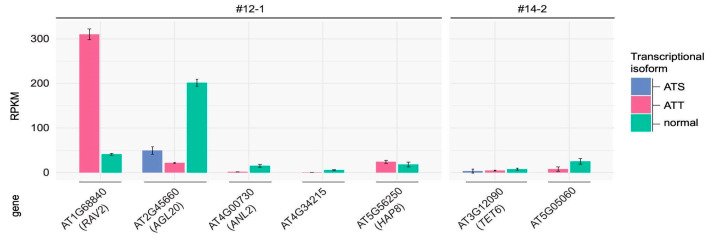
Differential isoform expression for genes with AS events caused by TEIs in the #12-1 and #14-2 plants. ATS (pink bars)—alternative transcription start; ATT (blue bars)—alternative transcription termination; normal (cyan bars)—wild-type transcript isoform.

**Figure 8 ijms-26-04816-f008:**
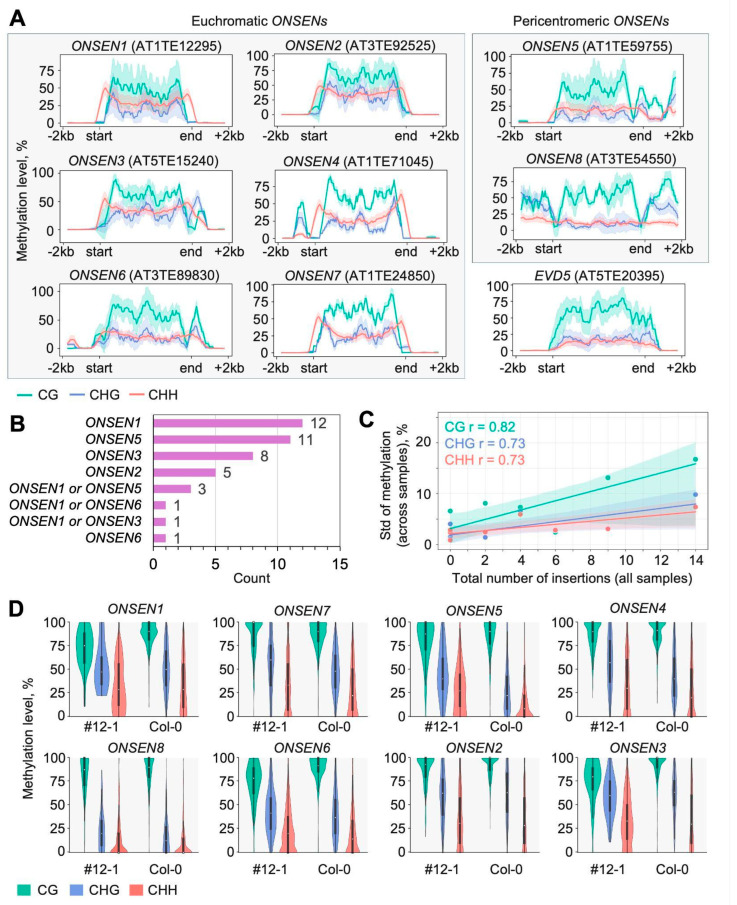
(**A**) Methylation patterns of the original *ONSEN* elements and *EVD5* retrotransposon (AT5TE20395). Metaplots show mean (continuous line) and standard deviation (colored area around mean) of binned methylation across all samples; smoothing (window = 10 bins) is applied. The last row reflects centromeric and pericentromeric TEs, whereas the first 2 contain TEs located in chromosome arms. (**B**) Total number of insertions produced by *ONSENs*. (**C**) Standard deviation of methylation levels increases with the number of insertions *ONSEN* produced. Dots represent original *ONSENs* with the corresponding number of insertions. The line represents the linear regression model fit, and the shadowed area is the confidence interval; r is the Pearson correlation coefficient. (**D**) *ONSEN* methylation levels in the #12-1 plant and Col-0.

**Figure 9 ijms-26-04816-f009:**
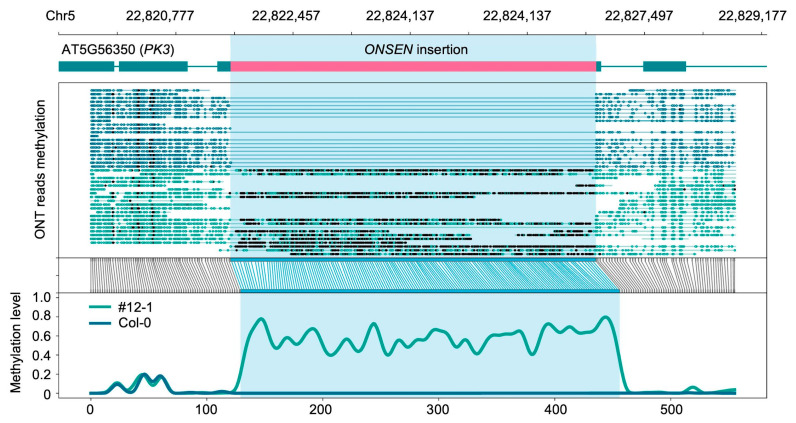
Methylation calls in the TEI region in Col-0 (blue track, no insertion) and #12-1 plant (green track, insertion is hypermethylated). Visualization made with methylartist (see Section 4 ). From top to bottom: *Arabidopsis* pseudoreference coordinates; gene annotation with *ONSEN* insertion; ONT read mapping with indicated methylated (black circles) and unmethylated (colored circles) cytosine; raw log-likelihood ratios; and smoothed methylated fraction plot.

**Figure 10 ijms-26-04816-f010:**
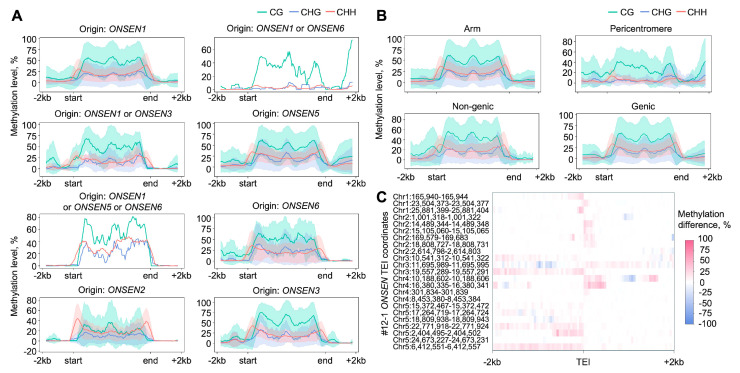
(**A**) Methylation patterns in novel *ONSEN* copies (TEIs) grouped by origin. Metaplots show mean (continuous line) and standard deviation (colored area around mean, if more than 1 insertion was identified) of binned methylation across all samples; smoothing (window = 10 bins) is applied. (**B**) Methylation patterns in novel *ONSEN* copies (TEIs) grouped by chromosomal location. Metaplots show mean (continuous line) and standard deviation (colored area around mean) of binned methylation across all samples; smoothing (window = 10 bins) is applied. (**C**) CHH methylation level difference between 12-1 TEI flanking regions and corresponding positions in Col-0. Mean binned (50 bp) combined methylation levels from two biological replicates are shown.

**Table 1 ijms-26-04816-t001:** Summary for *ONSEN* insertions in M2 plants according to the whole-genome nanopore sequencing data.

M2 Plant	New *ONSEN* Insertions	Intragenic Insertions, %	Homozygous Insertions	Heterozygous Insertions
#1-1	5	80.0	5	0
#3-6	2	100	0	2
#7-5	3	66.6	1	2
#12-1	32	90.6	18	14
#13-1	8	100.0	3	5
#14-2	6	100.0	2	4
Total	57	89.6	30	27

#—identity number of individual plant.

## Data Availability

The nanopore data produced for this study are available in the Sequence Read Archive (SRA), NCBI, under Bioproject Accession PRJNA1263850.

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
