# Peer review of "DNA Methylation and Alternative Splicing Safeguard Genome and Transcriptome After a Retrotransposition Burst in Arabidopsis thaliana"

_ijms, 2025, doi:10.3390/ijms26104816_

Round 1
Reviewer 1 Report
Comments and Suggestions for Authors
In this study, authors investigated the distribution of the insertion of ONSEN in Arabidopsis thaliana, and the influence of ONSEN insertion in epigenome and transcriptome. Overall, the authors performed comprehensive data analysis. However, some conclusions lack directly evidence. Some concerns are below.
Major concerns:
(1) While this study focuses on ONSEN insertions in A. thaliana, the introduction lacks critical context about ONSEN biology and other related studies, and the broader significance of investigating ONSEN specifically in A. thaliana is not well described.
(2) This study described that the ONSEN were mostly inserted in regions with enriched H3K27me3, H2A.Z, and H3K4me2. I wonder whether these epigenetic signals induce the insertion of ONSEN or whether the insertion of ONSEN alters the adjacent epigenetic signals.
(3) In this study, DNA methylation profile on genes with ONSEN insertions was associated with their chromosomal position, and the genes with ONSEN insertions share the similar DNA methylation patterns with genes inserted by EVD5. Does this mean that the methylation pattern of a gene is only related to its chromosomal position and not to whether or by which LTR is inserted?
(4) Authors described that RdDM pathway safeguarding the plant genome and transcriptome integrity after retrotransposition burst. However, there is a lack of siRNA sequencing data to directly support this conclusion.
Minor concerns:
(1) Figure 8C: The standard deviation of DNA methylation levels cannot exceed 0.5.
(2) Figure 8D: The y-axis is missing labels.
(3) Figure 8A, 10A, and 10B: The “%” is missing on the y-axis.
Author Response
Dear Reviewer,
We are grateful for your insightful feedback and the efforts you have made to enhance our manuscript. We have addressed all suggestions to improve the clarity and robustness of our study.
Detailed responses to the comments are provided below:
Major concerns:
Q1: While this study focuses on ONSEN insertions in A. thaliana, the introduction lacks critical context about ONSEN biology and other related studies, and the broader significance of investigating ONSEN specifically in A. thaliana is not well described.
A1: The corresponding paragraph has been added to the introduction.
Q2: This study described that the ONSEN were mostly inserted in regions with enriched H3K27me3, H2A.Z, and H3K4me2. I wonder whether these epigenetic signals induce the insertion of ONSEN or whether the insertion of ONSEN alters the adjacent epigenetic signals.
A2: While the precise mechanisms remain under investigation, ONSEN and other Ty1/copia retrotransposons (e.g., ATCOPIA93) are indeed guided by chromatin features such as the histone variant H2A.Z (https://doi.org/10.1038/s41467-019-11385-5). Our study leveraged publicly available chromatin data from the A. thaliana reference genome, which reflects the pre-existing epigenetic landscape prior to ONSEN insertions. While we demonstrated that new ONSEN insertions alter local DNA methylation patterns, whether this also leads to histone modification changes remains an open question for further exploration.
Q3: In this study, DNA methylation profile on genes with ONSEN insertions was associated with their chromosomal position, and the genes with ONSEN insertions share the similar DNA methylation patterns with genes inserted by EVD5. Does this mean that the methylation pattern of a gene is only related to its chromosomal position and not to whether or by which LTR is inserted?
A3: Our study demonstrates that chromosomal position-dependent DNA methylation patterns are specific to transposons (e.g., ONSEN retrotransposons and EVD DNA transposons), but not observed in protein-coding genes. DNA methylation of protein coding genes is regulated by different way and mainly occurs in CG contexts through MET1 maintenance methyltransferase. Notably, ONSEN insertions into genic regions did not significantly alter host gene methylation patterns.
Q4: Authors described that RdDM pathway safeguarding the plant genome and transcriptome integrity after retrotransposition burst. However, there is a lack of siRNA sequencing data to directly support this conclusion.
A4: Indeed the direct evidence of action of RdDM pathway requires siRNA data that we did not obtain for our plants. However, the previous reports provided multiple lines of evidence that ONSEN insertions are initially recognized by RdDM pathway [Ito et al. 2011; Matsunaga et al. 2015; Niu et al. 2022]. But we agree that the observed pattern of ONSEN insertion DNA methylation can be shaped by different pathways. Therefore, we change the title of the paper to ‘DNA Methylation and Alternative Splicing Safeguard genome and transcriptome after a retrotransposition burst in Arabidopsis thaliana’.
A5: Minor concerns:
The necessary changes have been made.
Reviewer 2 Report
Comments and Suggestions for Authors
This is a strong and well-designed manuscript that provides valuable insights into how Arabidopsis responds epigenetically and transcriptionally to new transposon insertions. The use of long-read sequencing, de novo genome assembly, and RdDM pathway analysis is commendable and adds mechanistic depth.
To further improve the manuscript, I suggest the following:
- Clarify Functional Consequences of Transcript Isoforms:
You describe the generation of intronized and truncated transcripts upon ONSEN insertion, but the downstream biological impact is not fully explored. Are these isoforms dominant, or do they coexist with wild-type transcripts? A brief functional annotation or discussion of possible protein-coding or regulatory roles would be helpful. - Improve Figure Clarity and Captioning:
Some figures may need clearer labeling. Please ensure that all tracks, color codes, and axes are fully defined within figure legends to improve standalone interpretability. - Enhance Integration of Results in Discussion:
The results are currently presented in well-separated blocks. The Discussion would benefit from a more integrated narrative that synthesizes how DNA methylation, splicing, and transcriptional changes work together to mitigate retrotransposon effects. - Minor Edits and Terminology Consistency:
Define abbreviations (e.g., ATT, ATS, RdDM) at first use.
Correct small typos (e.g., “ransposon display”).
Consider adding a summary schematic illustrating how RdDM and splicing contribute to transcriptome protection post-insertion.
Some references are not in ijms citation formatting. (e.g., “We noted that the most active original ONSENs, as well as its novel TEIs, have a high 475 level of methylation standard deviation, most pronounced in CG context. It is known that 476 chemical treatment with zebularine leads to DNA hypomethylation in all contexts [Grif-477 fin, 2016], which cannot be completely eliminated in the first generations [Huc, 2022; 478 Gerardo Del Toro, 2024]. In this case, some part of the lost symmetric methylation in the 479 TE body should be recovered de novo through RdDM pathway [Cuerda-Gil, 2016]. Alt-480 hough PolII-derived siRNAs can act in trans to lead to de novo methylation of all TE family 481 members, it is difficult to assess the impact of the sequence specificity of the siRNA pool 482 for the sequence to be methylated de novo [Slotkin, 2006].”)
Comments on the Quality of English LanguageDefine abbreviations (e.g., ATT, ATS, RdDM) at first use.
Correct small typos (e.g., “ransposon display”).
Consider adding a summary schematic illustrating how RdDM and splicing contribute to transcriptome protection post-insertion.
Some references are not in ijms citation formatting. (e.g., “We noted that the most active original ONSENs, as well as its novel TEIs, have a high 475 level of methylation standard deviation, most pronounced in CG context. It is known that 476 chemical treatment with zebularine leads to DNA hypomethylation in all contexts [Grif-477 fin, 2016], which cannot be completely eliminated in the first generations [Huc, 2022; 478 Gerardo Del Toro, 2024]. In this case, some part of the lost symmetric methylation in the 479 TE body should be recovered de novo through RdDM pathway [Cuerda-Gil, 2016]. Alt-480 hough PolII-derived siRNAs can act in trans to lead to de novo methylation of all TE family 481 members, it is difficult to assess the impact of the sequence specificity of the siRNA pool 482 for the sequence to be methylated de novo [Slotkin, 2006].”)
Author Response
Dear Reviewer,
We are grateful for your insightful feedback and the efforts you have made to enhance our manuscript. We have addressed all suggestions to improve the clarity and robustness of our study.
Detailed responses to the comments are provided below:
Q1: Clarify Functional Consequences of Transcript Isoforms:
You describe the generation of intronized and truncated transcripts upon ONSEN insertion, but the downstream biological impact is not fully explored. Are these isoforms dominant, or do they coexist with wild-type transcripts? A brief functional annotation or discussion of possible protein-coding or regulatory roles would be helpful.
A1: We thank the reviewer for this helpful suggestion, which has allowed us to strengthen the biological interpretation of our results. As we mentioned in the manuscript, among eight homozygous TEIs, three intronic and one exonic TEIs underwent intronization of full-length ONSEN insertions and these events does not change initial protein sequences. We also showed the relative representation of TE-genic and normal (wild-type) isoforms for genes with ONSEN insertions in plant #12-1 and #14-2 and included this information into Figure 7. As you can see, we observed significantly higher abundance of TE-genic isoforms only for RAV2 gene. To better describe these finding, we have now included a brief analysis of the potential implications at the proteomic level in the Results and Discussion sections, and added an illustration in the supplementary figures (now Figure S4).
Q2: Improve Figure Clarity and Captioning:
Some figures may need clearer labeling. Please ensure that all tracks, color codes, and axes are fully defined within figure legends to improve standalone interpretability.
A2: We thank the reviewer for this comment. The corresponding changes have been made in the MS.
Q3: Enhance Integration of Results in Discussion:
The results are currently presented in well-separated blocks. The Discussion would benefit from a more integrated narrative that synthesizes how DNA methylation, splicing, and transcriptional changes work together to mitigate retrotransposon effects.
A3: We agree with this comment. We did the following changes to the Discussion section:
- We modified the order of the main Discussion blocks.
- The subtitles reflecting main conclusions have been added.
- In the last section (‘3.4 Dual Defense: DNA Methylation and Alternative Splicing Rapidly Mitigate ONSEN Insertions in Arabidopsis’) the additional information about interplay between alternative splicing and DNA methylation has been added.
Q4: Minor Edits and Terminology Consistency:
The corresponding corrections have been made.
Round 2
Reviewer 1 Report
Comments and Suggestions for Authors
Accept in present form.